# Hepatocellular Carcinoma: The Role of MicroRNAs

**DOI:** 10.3390/biom12050645

**Published:** 2022-04-27

**Authors:** Sharad Khare, Tripti Khare, Raghu Ramanathan, Jamal A. Ibdah

**Affiliations:** 1Division of Gastroenterology and Hepatology, Department of Medicine, University of Missouri, Columbia, MO 65212, USA; khares@health.missouri.edu (S.K.); kharet@health.missouri.edu (T.K.); raghu.ramanathan@health.missouri.edu (R.R.); 2Harry S. Truman Veterans Hospital, Columbia, MO 65201, USA; 3Department of Medical Pharmacology and Physiology, University of Missouri, Columbia, MO 65212, USA

**Keywords:** hepatocellular carcinoma, microRNA, metastasis, heterogeneity, therapeutics

## Abstract

Hepatocellular carcinoma (HCC) is the second leading cause of cancer-related deaths worldwide. HCC is diagnosed in its advanced stage when limited treatment options are available. Substantial morphologic, genetic and epigenetic heterogeneity has been reported in HCC, which poses a challenge for the development of a targeted therapy. In this review, we discuss the role and involvement of several microRNAs (miRs) in the heterogeneity and metastasis of hepatocellular carcinoma with a special emphasis on their possible role as a diagnostic and prognostic tool in the risk prediction, early detection, and treatment of hepatocellular carcinoma.

## 1. Introduction

Hepatocellular carcinoma (HCC) is the most common and deadly liver cancer and the second leading cause of cancer-related deaths worldwide [1,2]. HCC, aggressive in nature, accounts for 90% of primary liver cancers [3]. HCC normally develops in the setting of cirrhosis and the process of tumorigenesis is further promoted by chronic viral hepatitis related to the hepatitis B (HBV) and hepatitis C viruses (HCV), alcohol-induced injury, non-alcoholic fatty liver disease, exposure to aflatoxins or genetic disposition [4]. Diet-induced HCC is an emerging problem in developed as well as in developing countries [5,6]. Over the past 15 years, reported cases of HCC have more than doubled due to late diagnosis [7] with limited treatment options and marginal clinical benefits to patients.

Among the limited potential curative strategies are liver resection and liver transplantation as well as loco-regional therapies, such as radiofrequency ablation and transarterial chemoembolization [8]. Because of shortage of healthy livers, the transplants are provided crucially to patients that have the best chance of long-term survival. Cytotoxic systemic therapy is limited by tumor chemoresistance and patient intolerance [9].

Multiple molecular regulations set a precedence for the high rate of recurrence and may lead to tumor heterogeneity in HCC. In this regard, substantial heterogeneity has been reported between the multiple tumor foci in a single patient. Therefore, more reliable biomarkers are needed for the diagnosis, treatment and surveillance of HCC [8,10].

## 2. Heterogeneity in HCC

Cancer heterogeneity has been recognized as an important clinical determinant of patient outcomes, such as response or resistance to anti-cancer therapies [11,12]. Heterogeneity is prevalent in cancer, both between and within individuals. Multiple morphologic (differentiation status and cytologic features), genetic (mutational background), epigenetic (DNA methylation) and microenvironment (hypoxia gradients and local oxidative stress) variations create heterogeneity amongst different tumors [10,13]. HCC is a highly heterogenous cancer with significant intratumor heterogeneity. The pathologic classification of HCC is based on the degree of cellular differentiation. In a single tumor, the cancerous tissue of two different histological grades may be present. In addition, tissue obtained from HCC may exhibit different immunohistochemical characteristics in the same tumor. Furthermore, genetic heterogeneity has been described in HCC. Thus, HCC is less likely to be caused by a single driver mutation. The intratumor heterogeneity of HCC plays an important role in the prognosis of the disease. Hence, the detection of HCC intratumor heterogeneity is important for the development of effective targeted therapies. Though liver transplantation, surgical resection and radiofrequency ablation (RFA) offer a curative treatment for HCC, they are not an option for patients with intermediate/advanced stage HCC. Sorafenib, a multikinase inhibitor of several tyrosine protein kinases, is implicated in the treatment of patients with intermediate/advanced HCC. It has shown a modest increase in median survival in clinical trials [14]. However, the adequate concentration of this drug is not achieved because of vascular heterogeneity within the tumor, resulting in a reduced response to therapy. Thus, intratumor heterogeneity also plays a role in drug resistance. Therefore, the better understanding of the intratumor heterogeneity of HCC should provide critical knowledge about the prognosis of the disease and response to potential future therapy.

### 2.1. Morphologic Heterogeneity in HCC

HCC can be classified pathologically on the basis of the degree of cellular differentiation, which include well differentiated to moderately and poorly differentiated and undifferentiated tumors. The two histological grades can be present in one tumor, thereby exhibiting different immunohistochemical characteristics in the same tumor. Apart from differentiation, intratumor heterogeneity also decides the size of the tumor and lymphovascular spread. Some of the histochemical markers used for HCC diagnosis include pCEA, CD10, alpha fetoprotein (AFP), hepatocyte paraffin1 (hepPar1), cytoplasmic thyroid transcription factor-1 (TTF1), glutamine synthetase, GPC3, CK8 and CK18, but unfortunately none of them is specific of early stage HCC. AFP is most insensitive of all because it is not expressed by all HCC cells [15]. Zhang Q et al. recently proposed an immunophenotypic classification of HCC using two markers (CD45 and Foxp3) that will facilitate prognostic prediction and decision making for the choice of therapies [16].

HCC is well known for morphologic intratumor heterogeneity, but very few systematic analyses of this phenomenon have been performed. Intratumor heterogeneity is detectable in the majority of HCC cases (87%), with 26% of cases at the level of morphology. Further, 39% of cases are classified at the combined morphologic and immunohistochemical level and 22% at the combined morphologic and immunohistochemical treatment target level with known mutational status, for example, TP53 and β-catenin [4].

Intratumor heterogeneity poses a challenge for the development of a robust HCC classification as well as a targeted therapy and may contribute to treatment failure and drug resistance in many cases of HCC. HCC is known to frequently display heterogeneous growth patterns and/or cytologic features within the same tumor. This kind of plasticity of phenotypes or intratumor heterogeneity has already been described in several solid tumors, from skin, breast and kidney diseases [17,18,19,20,21]. In small HCC, measuring 3 to 5 cm in diameter, up to 64% of cases display intratumor heterogeneity at the level of histologic differentiation grade and proliferative activity, whereas in HCC smaller than 2 cm, intratumor heterogeneity is 25–47% [22,23]. In larger HCC, on the other hand, the true extent of intratumor heterogeneity with respect to morphologic, immunohistochemical and molecular features has not been systematically assessed [4].

### 2.2. Genetic Heterogeneity in HCC

Nault and Villanueva identified TERT promoter mutations with an overall frequency of 60%, as the most frequent somatic genetic defect in HCC [24]. It is also recurrently mutated in precancerous nodules as the earliest genetic alteration involved in the malignant transformation of HCC and possibly considered as a tumor “gatekeeper”. They speculated that TERT promoter mutations are present in the common ancestor cell and transmitted to its progeny and, therefore, are present in most tumor cells. Further studies are still needed to decipher HCC intratumor genetic heterogeneity using unbiased approaches, such as whole-exome or whole-genome sequencing, preferable by ultra-deep sequencing [13,25].

Large and independent studies have validated several molecular prognostic signatures derived from the tumor, despite tumor heterogeneity [26,27]. All of the biomarker-based molecular therapies approved in solid malignancies also did not account for genetic heterogeneity and are mainly based on traditional genetic analysis, for example, vemurafenib for BRAF V600E-mutated melanoma, cetuximab for wild-type RAS colorectal cancer and orcrizotinib for ALK-translocated non-small cell lung cancer [28].

Within-patient heterogeneity in HCC is well studied because it often presents with multiple tumor foci. In patients with multifocal HCC, the individual lesions usually arise from either the local dissemination of the primary tumor or from the oncogenic predisposition of the diseased liver. In the latter case, a patient with multifocal HCC may have multiple tumors presumably because of distinct genomic profiles and clonal unrelation, a situation that poses a significant challenge to genomic analyses. A microRNA biomarker of HCC recurrence, following liver transplantation, accounting for within-patient heterogeneity has been described [8].

## 3. MicroRNAs and Heterogeneity in HCC

Altered genomic and transcriptional landscapes are associated with carcinogenesis and include protein-coding genes as well as several classes of structurally and functionally different noncoding RNAs. At least 90% of the human genome is transcribed into noncoding RNAs that do not translate into proteins. Amongst the noncoding RNAs, miRs are a highly conserved class of small noncoding RNAs of ~18–24 nucleotides in length, which are involved in the regulation of gene expression at the transcription level by binding to their target gene promoters [29,30] and post-transcriptional level by degrading their target mRNAs and/or inhibiting their translation by binding to the 3′-untranslated region (3′UTR) of target mRNAs [31,32]. A single miR may regulate hundreds of target mRNAs with the same short recognition region; simultaneously, the 3′-UTR of most mRNAs has more than one binding site for different miRs. Since the discovery of the first miR lin-4 in 1993, more than 2000 miRs, to date, have been discovered in humans [33] that regulate one third of the genes in genome [34,35].

miRs are differentially expressed in all cancers as oncogenic miRs are generally enriched, while tumor-suppressor miRs are downregulated [36]. MiRs in HCC progression and metastasis regulate proliferation, apoptosis, invasion, the epithelial–mesesnchymal transition (EMT), angiogenesis, drug resistance and autophagy [37,38,39,40]. Amongst the tumor-suppressor miRs, some are involved in tumor initiation and progression [41,42,43], while others are involved in proliferation, invasion, metastasis and recurrence [44,45,46,47,48,49,50,51]. Similarly, some oncogenic miRs are associated with tumor development and progression [52,53,54,55,56,57,58,59,60], while others are associated with recurrence and metastasis [61,62,63,64,65,66] (Table 1).

MiR-122, a tumor-suppressing liver-specific miR, accounts for 70% of the total miR in hepatic tissues. It modulates hepatic lipid metabolism and inhibits viral replication in HBV-related HCC by targeting NDRG3, a member of N-myc downstream-regulated gene family. Both miR-122 and NDRG3 are considered possible therapeutic targets for HBV-related HCC [92]. MiR-122 is downregulated in HCC with a cirrhotic background as well as with other tumor-suppressor miR let-7 family members, miR-221, and miR-145 [93]. Studies have shown that miR associated with cell-cycle inhibition (miR-34a, miR-101, miR-199-a-5p and miR-223) [68,77,94,95] were downregulated, whereas the ones associated with cell proliferation and the inhibition of apoptosis (miR-17-92 polycistron, miR-21, miR-96, miR-221 and miR-2240) are upregulated in HCC [96,97,98,99]. Studies have also reported the association of miR-221 with the multifocality of tumors [100]. Jovel et al. described heterogeneity in terms of the upregulation of let-7i-5p, miR941, miR-301b, miR-210-3p and miR-21-3p, and the downregulation of miR-483-3p, miR-429, miR200a-3p, miR200b-3p, miR-125a-5p, miR-203a and miR-335-3p in liver-transplant recipients and the upregulation of miR1269a, miR-10b-5p, miR-217 and miR-452-5p and the downregulation of miR-483-3p, mir-483-5p, miR-4485-3p and miR-214-3p in liver resection patients [10].

The differential expression level of miRs in HCC regulates apoptosis and the mTOR, Wnt, JAK/STAT and MAPK pathways [101]. Many studies have shown that the downregulation of tumor-suppressor or the upregulation of oncogenic miRs results in the activation of the mTOR pathway in HCC. The tumor-suppressor miRs involved in the suppression of the mTOR pathway are miR758-3p, miR-142, miR-199b-5p, miR-187, miR-497, miR-99a, miR-592, miR-296-5p, miR-139-5p, miR-15b-5p, miR-345 and miR-223 [102,103,104,105,106,107,108,109,110,111,112,113]. Conversely, the oncogenic miRs, involved in the activation of the mTOR pathway, include miR-33a, miR-302d, miR-23b, miR-181a, miR-155-5p and miR-25 [114,115,116,117,118,119].

On the basis of a miRNome study in HCC tissues, several important deregulated miRs have been suggested, such as miR-361 [78], miR-122 [74] and miR-199 [75]. However, as more than one miR is deregulated in cancer cells and a single miR can target multiple mRNAs, it is an ongoing process to identify the deregulated miRs and uncover their roles in HCC development. MiR-500a, which is upregulated in HCC tissues, targets the BH3-interacting death agonist (BID) protein and can also serve as a possible prognostic predictor and therapeutic target in HCC patients [91]. Wu H. et al. demonstrated that miR-206 is a robust tumor suppressor and strongly prevented the development of HCC in AKT/Ras and cMyc HCC mouse models [76]. Studies by Sun J.J. et al. demonstrated that miR-361-5p inhibits cancer cell growth by targeting CXCR6 in HCC. The knockdown of CXCR6 and the forced expression of miR-361-5p inhibited tumor growth both in vitro and in vivo. It, therefore, serves as a tumor suppressor and might serve as a novel therapeutic target for the treatment of HCC patients [78].

## 4. MicroRNAs in Metastasis of HCC

Increasing data suggest a role of miRs in liver development, regeneration and metabolism, in various liver diseases, including liver hepatitis, steatosis, cirrhosis and HCC [120]. The role of several miRs in human cancer onset and progression, including invasion and metastasis, has been demonstrated [71]. For the initiation and maintenance of a mobile phenotype, the responsible cellular mechanisms include the cessation of cell polarity, cytoskeletal reorganization, re-connection with the microenvironment and the activation of pro-migratory intracellular molecules. However, the upstream regulation, leading to a highly invasive cellular phenotype, is not completely understood. Recently, Chuang at al illustrated that the dysregulation of a single miR, miR-494, supports HCC invasiveness through the epigenetic regulation of a miR network [121]. Interestingly, the role of miR-494 is both tumor suppressing and oncogenic across different tumor types and HCC, which reflects its pleiotropic character targeting distinct mRNA sets according to the genomic background and microenvironment. Its target in HCC involves the tumor-suppressor gene PTEN and MCC [1].

In recent decades, several miRs have been associated with HCC progression and metastasis, for example, miR-148a [122], miR-124 and miR-203 [69], miR-138 [123], miR-122 [124] and miR-30a-5p [125]. However, there are many more miRs that play a role in the progression and metastasis of HCC. For example, miR-141-3p inhibits the progression and metastasis of HCC by inhibiting EMT through the targeting of the Golgi protein 73 (GP73). It induces the expression of E-cadherin (epithelial cell marker), occludin (a marker of tight junctions) and cytokeratin 18 (CK 18) (a noninvasive cell marker), but reduces the expression of two mesenchymal markers N-cadherin and vimentin [72]. GP73 restores the inhibitory effects of miR-141-3p on the invasion and metastasis of HCC cells. MiR-487a, on the other hand, promotes the proliferation and metastasis of HCC by binding to phosphoinositide-3-Kinase regulatory subunit 1 (PIK3R1) and Sprouty-related EVH1 domain containing 2 (SPRED2) [90]. MiR-874 negatively regulates δ opioid receptor (DOR), which can suppress the proliferation and metastasis in HCC tumor by targeting the DOR/EGFR/ERK pathway [80], whereas miR-501-3p controls the metastatic process of HCCs by targeting Lin-7 homolog A (LIN7A) [79]. Additionally, miR-219-5p promotes HCC cell proliferation, invasion and metastasis in nude mice models bearing human HCC tumors by targeting the cadherin 1 (CDH1) gene [87]. MiR-197, which is dysregulated in several cancers, including lung, breast, ovarian, colorectal, thyroid, prostate, head and neck carcinoma, HCC as well as in non-alcoholic fatty liver disease, plays an important role in EMT. It promotes the invasion and metastasis of HCC cells by activating Wnt/β-catenin signaling by targeting Axin-2, Naked cuticle 1 (NKD1) and Dickkopf-related protein 2 (DKK2) [86]. However, miR-197-3p, which is downregulated in HCC tissues, inhibits the metastasis of HCC cells both in vitro and in vivo. Its novel target in HCC cells is the zinc finger protein interacting with K protein 1 (ZIK1) [73]. MiR-424-5p, which is involved in the progression, invasion and intrahepatic metastasis of HCC regulates Tripartite motif-containing 29 (TRIM 29), a member of the TRIM protein family that participates in the formation of nucleic-acid-bound homodimers or heterodimers, acting as transcriptional regulators of carcinogenesis and differentiation. MiR-221 and 222 also promote metastasis in HCC by targeting Plant homeodomain finger 2 (PHF2), AKT pathway, PTEN, CDK inhibitor p27 and DDIT4 [88,89]. Figure 1 depicts the reported miRs associated with HCC heterogeneity as well as metastasis. Interestingly, four reported miRs (MiR 221, miR 21, miR 203 and miR 214) are common to both heterogeneity and metastasis in HCC (Figure 1). All these miRs can serve as possible diagnostic and prognostic markers for HCC.

## 5. Diagnostic and Prognostic MicroRNAs in HCC

The survival rate of HCC is at most 5 years, which is still very low, partly because of the unsatisfactory results of conventional biomarkers (e.g., DPC, AFP and AFP-L3) that are often unable to distinguish between cancer and inflammatory diseases, such as chronic hepatitis or liver cirrhosis [126]. On the other hand, miRs have a high specificity in cancer detection and classification. They are highly stable and can be accurately detected under extreme conditions in a wide variety of body fluids [127,128]. The dysregulation of miRs is considered an early event in tumorigenesis, so miRs are promising biomarkers for the early diagnosis of cancer [129,130,131]. However, variations in the isolation protocols, cohort specifications, detection platforms and tumor heterogeneity often result in poor consensus regarding circulating miR profiles in patients with HCC [132].

In cancer, miRs have shown promise as both diagnostic and prognostic biomarkers [133]. A recent study reported miR-718 from serum exosome samples serves as biomarker of HCC recurrence after liver transplantation [134]. Exosomes are a class of extracellular vesicles derived from most cell types and are present in biological fluids, such as serum, plasma, urine, saliva, ascites and cerebrospinal and amniotic fluids. Studies reported their role in mediating cell-to-cell communication. Several functions of exosomes have been characterized, including cellular proliferation, differentiation, apoptosis, angiogenesis and immune regulation. Exosomes exhibit these functions by interacting with the surface receptors of recipient cells, thus transmitting biomolecule miRNAs. Exosome miRs have the potential to be used as biomarkers for HCC diagnosis and prognosis. Sohn et al. used fluorescent quantitative PCR to detect the expression levels of serum exosomal miRs in patients with chronic hepatitis B, liver cirrhosis and HCC. They discovered that the serum levels of exo-miR-18a, exo-miR-221, exo-miR-222 and exo-miR-224 in patients with HCC were significantly higher than those in patients with chronic hepatitis B or liver, leading to the conclusion that serum exosomal miRs can be employed as novel biomarkers for HCC screening and diagnosis [135]. Patients with serum exo-miR-215-5p overexpression had a significantly lower disease-free survival than patients with low serum exo-miR-215-5p expression, according to a Kaplan–Meier analysis. Simultaneously, the expression level of exo-miR-215-5p rises with the progression of the tumor stage and can be employed as a predictive biomarker in HCC [136]. Another study indicated that, as compared to patients with liver cirrhosis, exo-miR-21 and exo-miR-96 expression levels in HCC patients’ exosomes and plasma were significantly higher, while exo-miR-122 expression was significantly lower. Exo-miR-122, exo-miR-21 and exo-miR-96 are substantially more accurate in the diagnosis of HCC in diverse populations than plasma microRNA and AFP levels, and are prospective biomarkers for the early identification of HCC [137]. Some exosomal miRs have recently been identified as recurrence-specific indicators, particularly in HCC patients. Exo-miR-92b was more expressed in patients with recurrence after surgery than in patients without recurrence, and it can be employed as a useful biomarker for predicting the probability of HCC recurrence [138,139]. Other studies have proposed miRs as biomarkers of HCC recurrence from solid tumor biopsies based on their miR expression profiles [67,70,140,141]. Yang et al. did a meta-analysis of miR expression in HCC and identified a meta-signature of five upregulated (miR-221, miR-222, miR-93, miR-21 and miR-224) and four downregulated (miR-130a, miR-195, miR-199a and miR-375) miRs. These nine miRs are associated with cell signaling and cancer pathogenesis and could serve as potential diagnostic and therapeutic targets of this malignancy [142]. Table 2 lists diagnostic and prognostic miRs in HCC.

In HCV-induced HCC, miR-1269, miR-224, miR-224-3p and miR-452 are upregulated, whereas miR-199a-5p, miR-199a-3p and miR-199b are downregulated as compared to healthy controls, HCV-induced cirrhosis and HBV-induced liver failure [143]. Furthermore, miR-122, miR-199a and miR-16 have been established as potential biomarkers of HCV-induced HCC in Egyptian patients [144]. Li et al. identified a 13-miR panel (miR-375, miR-92a, miR-10a, miR-223, miR-423, miR-23b, miR-23a, miR-342-3p, miR-99a, miR-122a, miR-125b, miR-150 and let-7c) as a novel noninvasive biomarker in HBV-mediated HCC and this panel has made possible the diagnosis and differentiation of HBV-induced HCC cases from healthy controls, HCV and subjects with HBV infection without HCC [146]. Recently, a panel of seven miRs (miR-29a, miR-29c, miR-133a, miR-143, miR-145, miR-192 and miR-505) is able to differentiate HCC patients from healthy volunteers, patients with cirrhosis and patients with chronic HBV infection [128,155]. Similarly, other studies represent the combination of AFP and a panel of three miRs (miR-92-3p, miR-107 and miR-3126-5p) as an effective diagnostic aid for early-stage and low-level AFP-HCC patients [147]. The overexpression of an eight-miR panel (miR-20a-5p, miR-25-3p, miR-30a-5p, miR-92a-3p, miR-132-3p, miR-185-5p, miR-320a and miR-324-3p) can be used to differentiate between HBV-positive cancer-free controls and HBV-positive HCC patients [148].

The use of the biomarkers for epigenetic changes involving miR for the early detection and risk prediction of HCC [156] and as prognostic or diagnostic markers in the clinical management of patients with HCC is a very promising area [157]. For example, miR-21 and miR-199a are potential biomarkers for HCC [149] and a panel of miRs (miR-192-5p, miR-21-5p and miR-375, alone or combined with AFP) may serve as a blood-based early detection biomarker for HCC screening. Circulating miR-21 is characterized as potential diagnostic biomarker for HCC because of some of its unique advantages over others. MiRs offer the advantages of being minimally invasive, the serum levels being stable and reproducible, and levels not being influenced by both cirrhosis and viral status with a significant overexpression even in early stage HCC patients. MiRs can serve as novel co-biomarkers to AFP to improve the diagnostic accuracy of early stage HCC [158]. Sorafenib administration has been reported to modulate the expression of miRs. Fourteen miRs are upregulated by Sorafenib treatment in HCC cell lines [159]. The overexpression of miR-122 in HCC cell lines makes them sensitive to Sorafenib treatment [160] and the overexpression of miR-122 in HCC cells makes them sensitive to doxorubicin treatment [161]. However, the decreased expression of miR-34a indicates the resistance of HCC cells to Sorafenib [162]. Recently, an artificial lncRNA was generated that overcomes the Sorafenib resistance of HCC cells by targeting multiple miRs [163].

MetastamiRs are miRs that promote or suppress the migration and metastasis of cancer cells, thereby exhibiting significant functional correlation with the prognosis of HCC. Unlike targeted therapy, metastamiRs have been shown to target multiple mRNAs and signaling pathways with the considerable suppression of cancer metastasis that might in future enable an anti-HCC miR drug development [164]. In addition to miRs, its upstream regulators and downstream target genes can also be used as alternative biomarkers and therapeutic targets for the diagnosis and therapy of HCC. Various miR target prediction software, such as MiRanda 3.0, TargetScan5.1 and miRecords, can be used to study miR targets and these targets can be analyzed further by gene ontology hierarchy (http://pantherdb.org, or https://david.ncifcrf.gov/ (accessed on 30 September 2021) [165]. RNA-seq and miR array analysis can add more miRs involved in HCC development and some of the specific regulated miRs can be used in targeted therapy. Instead of targeting specific miRs, signaling pathways involved in cancer development can also be used for precision treatment by miR and the possible method used for miR delivery include the sleeping beauty transposon via hydrodynamic tail vein injection [166]. The CRSIPR/CAS9 genome editor method and liver-specific gene knockout or knock-in mice are some of the approaches used for miRs and their target gene functions in HCC [167].

Although treatment options for patients with advanced HCC have improved in recent years, it is critical to develop prognostic markers that anticipate tumor growth and worsening liver function in order to move patients to more successful treatment lines [168,169]. The most important finding from Frundt et al. suggest that exosomal miR-192 levels in the plasma have a diagnostic and predictive value in an HCC patient cohort, and that exosomal miR-192 presence was linked to a lower overall survival rate (OS). Exosomal miR-192 levels were found to be higher in the blood of HCC patients by Xue et al. [150]. Furthermore, the high serum levels of exosome- and cell-free circulating miR-192 were linked to poor OS, according to Zhu et al. [170]. Because these patients were treated with surgical resection, microwave ablation (MWA) or transarterial chemoembolization (TACE), the enrichment of miR-192 in exosomes can offer predictive value, especially for patients at an early or intermediated tumor stage. Suheiro et al. recently found that changes in exosomal miR-122 expression are linked to survival in HCC patients treated with TACE, demonstrating miRNAs’ ability to act as biomarkers for therapeutic monitoring [171]. miR-16 is known to be downregulated in HCC cells, and overexpression suppresses HCC cell proliferation, invasion and metastasis [172], implying that miR-16 functions as a tumor suppressor. miR-221 is an oncogenic miRNA that regulates the PTEN/PI3K/AKT and JAK-STAT3 signaling pathways, which are important in the development of HCC [173,174]. Exosomal miR-221 levels were found to be greater in HCC patients than in liver cirrhosis patients by Sohn et al. [135]. These findings point to miR-221’s potential utility as a tumor marker for HCC screening in individuals with hepatic cirrhosis. Zhang et al. reported that miRs, such hsa-miR-139-3p, hsa-miR-760 and hsa-miR-7-5p, have independent prognostic relevance, and were found to be strongly linked with HCC patients’ overall survival [175]. The above studies show that miRNAs are also predictive markers in patients with liver cirrhosis, which could aid assessment in these individuals.

## 6. Strategies for MicroRNA Potential Use in HCC as Therapeutic Targets

The main challenge faced by miR-based therapy is to reach the required drug levels in the tumor. However, they can be achieved with the chemical modification of therapeutic miRs [176]. Table 3 summarizes some of the miRs used in HCC therapeutics. Two strategies are involved in use of miR for cancer therapeutics. The first strategy involves inhibiting oncogenic miRs (OncomiRs) to gain function using miR antagonists, such as locked nucleic acids (LNA), antagomiRs and antimiRs. The most commonly used miR inhibitors are LNAs and antisense oligonucleotides [177]. LNAs are RNA analogs with very high affinity and specificity for complimentary miR. LNAs can be used in low doses and are more resistant to digestion by nucleases [178]. For example, the use of LNAs specific for miR-122 in non-human primates chronically infected with HCV suppressed long-term viral growth, supporting its use as therapeutic agent in HCC [179]. Further, the antagonist of miR-122, miravirsen, is used in a multi-center phase IIA trial in HCC patients and exhibited sequestration of mature miRNA and reduction in viral load [180]. Additionally, the use of Morpholino-anti-miR 487a oligomers effectively silenced miR-487a in mouse models, resulting in the inhibition of HCC tumor progression with no toxicity to mice in terms of weight loss, other visible impairments and animal death [90,181,182].

The second strategy involved in use of miR is replacement by re-introducing tumor-suppressor miRs to restore the loss of function [183,184,185,186,187,188,189,190,191,192,193,194,195]. The replacement miRs are either short double-stranded oligonucleotides or miR mimics, which are double-stranded RNA molecules with inverted bases and alkyl groups [196]. For example, the use of oligonucleotides in a pre-clinical study targeting miR-221 in an orthotopic HCC mouse model resulted in the inhibition of cell transformation and improved survival. Similarly, the AAV-mediated delivery of miR-122, miR-26a and miR-199a and the systemic restoration of miR-124, miR-29 and miR-375 (2′*O*-methyl-modified and cholesterol-conjugated form) could inhibit tumorigenesis in HCC animal models [179,197]. Additionally, miR mimics have been used successfully as a strategy. For example, a miRNA mimic of miR34 (MRX34) has been used in HCC patients in a phase 1 clinical trial. Despite the fact that Mirna Therapeutics terminated the trial early due to substantial immune-mediated side effects that resulted in four patient deaths, the dose-dependent regulation of key target genes shows that a miRNA-based cancer therapy can be effective. MiRs are functional in diverse cellular events and because of these properties, several clinical trials in cancer research utilizing miRs are currently underway (Available online: http://ClinicalTrials.gov (accessed on 15 August 2021)). Many studies show that miRNA-based treatments in cancer provide a proof-of-concept; however, this class of medications still needs to be researched further to prevent immune-related toxicity in patients.

## 7. Conclusions

HCC is a complex disease with the involvement of a variety of risk factors and usually diagnosed when cancer is in advanced stage with poor survival, frequent recurrence and limited therapy. Tumor heterogeneity, both at the clinical and molecular level, is well known in HCC and poses a challenge for the development of a targeted therapy. The lack of specific diagnostic markers for HCC presents challenges for the early detection of the disease and cancer therapy. There is an urgent need of novel diagnostic biomarkers to achieve the risk stratification and earlier diagnosis of HCC. MiRs are endogenous transcriptional and post-transcriptional regulators of gene expression and have a critical role in pathogenesis of HCC. They are expressed differentially even at very early stages of cancer and are involved in cancer heterogeneity and metastasis. The emerging role of miRs as novel clinical biomarkers is definitely going to change the face of HCC clinical evaluation through risk prediction, early diagnosis and determining the appropriate therapeutic course of action.

## Figures and Tables

**Figure 1 biomolecules-12-00645-f001:**
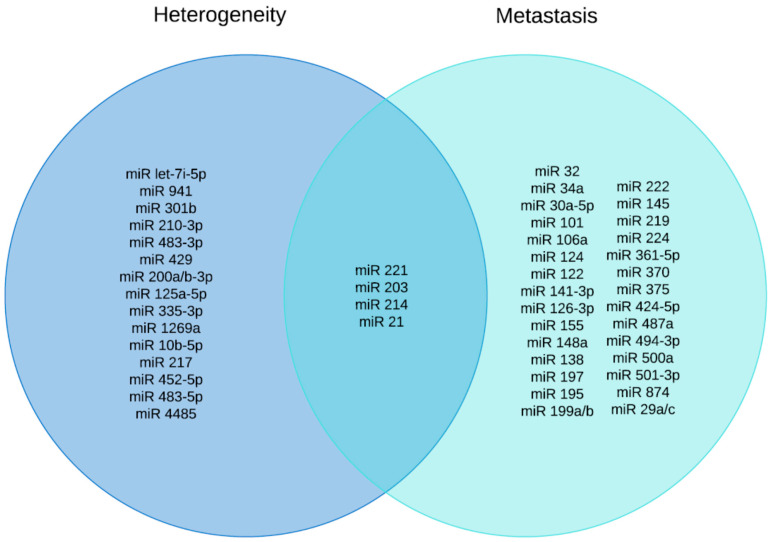
Venn diagram showing a list of microRNAs involved in HCC heterogeneity (19) and metastasis (33). Overlapping region shows microRNAs (4) that are common to both.

**Table 1 biomolecules-12-00645-t001:** Differentially expressed miRs in HCC.

	MicroRNAs	Cellular Target/Mechanism	Related To	Reference
**Tumor-Suppressor miRNAs**	miR-26a	FBXO11, Cyclin D2, Cyclin E2	Tumor initiation/ progression	[41]
	miR-17, miR19a, miR-20a, miR-24, miR-27a, miR-106a, miR-205, miR886-5p	E2F1, c-Myc, PTEN, SOX4,l c-Myc, E2F2, MMP14, FLT1 APC, CYP1B1, HNF3α, STAT3, MMP7 ZEB2, Bmi, ABL1, CAV1, K-RAS, Bax	Recurrence after liver Transplantation	[67]
	miR-29 a-c	SETDB1	Metastasis	[50]
	miR-34a	p53, EMT-transcription factor, Notch/Wnt/TGF-β/SMAD signaling	Progression/EMT	[43]
	miR-101	COX-2, Mcl-1	Proliferation/Migration/ Recurrence after transplant	[68]
	miR-122	WNT1, Bcl-W, p53	Proliferation/Metastasis/ Recurrence after transplant	[44] [67]
	miR-124	CDK6, VIM, SMYD3, IQGAP1	Metastasis	[69]
	miR-125a/b	SIRT7, LIN28B, mcl-1, IL-6R, Bcl2	Tumor differentiation	[70]
	miR-126-3p	Focal adhesion and MAPK signaling	Invasion/Metastasis/ Recurrence after transplant	[71] [67]
	miR-141-3p	GP73, E-cadherin, N-cadherin, occludin, vimentin and cytokeratin	Tumor growth/ Metastasis	[72]
	miR-145	IRS1, AKT/FOXO1	Proliferation/Migration/ Invasion	[49]
	miR-147	HOXC6	Recurrence after transplant	[67]
	miR-195	Cyclin D1, CDK6, E2F3, IKKα, TGF-beta-activated kinase1, MAP3K7 binding protein 3 (TAb3), NF-κB and PCMT1	Proliferation/ Tumorigenesis/Metastasis	[45]
	miR-197-3p	ZIK1	Metastasis	[73]
	miR199a/b-3p	PAK4/Raf/MEK/ERK pathway	Progression	[74]
	miR199a/b-5p	ROCK1/MLC and PI3K/Akt signaling	Proliferation/Invasion/Migration	[75]
	miR-203	CDK6,VIM, SMYD3, IQGAP1 and ABCE1	Epigenetic control of HCC/ Recurrence after transplant	[69]
	miR-206	cMET, CCND1 and CDK6	Growth of HCC	[76]
	miR-214	EZH2, β-catenin	Recurrence/Progression/ Metastasis	[51]
	miR-223	STMN1	Proliferation/HCC Development/Recurrence after transplant	[77]
	miR-302b	AKT2	Tumor growth	[42]
	miR-339	ZNF689	Proliferation/Invasion	[78]
	miR-15a, miR16 and miR-107	WNT3A	Proliferation/Cell cycle/Invasion	[48]
	miR-375	YAP and AEG-1	Proliferation/Migration/ Invasion	[47]
	miR-424-5p	TRIM 29	Proliferation/Invasion	[46]
	miR-501-3p	LIN7A	Progression/Metastasis	[79]
	miR-874	DOR, EGFR and ERK	Proliferation/Metastasis	[80]
	miR-140	MMP9	Invastion/Metastasis	[81]
	miR-193b	MAPK1	Invasion/Migration	[82]
	miR-377-3p	CELF1	Proliferation/Migration	[83]
	miR-605-3p	TRAF6	Metastasis/EMT	[84]
	miR-544b	BTG2	Proliferation/Migration/Invasion	[85]
**Oncogenic** **miRNAs**				
	miR-25	Bim	Development/Progression/ Poor prognosis	[57]
	miR-17-5p	PTEN, GalNT7, Vimentin	HCV-induced HCC	[56]
	miR-18a	ESR1	Proliferation	[53]
	miR-32	PTEN	Proliferation/Migration/ Invasion	[65]
	miR-92b	SMAD7	HCV-induced HCC	[60]
	miR-93	PTEN, CDKN1A, c-Met/p13k/AKT pathway	Development/ Progression	[55]
	miR-101	FOS oncogene	Viral HCC	[52]
	miR-106a		Tumor recurrence	[61]
	miR-106b	EMT	Migration/Metastasis	[66]
	miR-130b	PPAR-γ, TP53INP1	Tumorigenesis	[54]
	miR-155	c-Myc, C/EBPβ, APC, β-catenin, cyclin D1, TP53INP1	Invasion/Tumorigenesis/ Recurrence after transplant	[62]
	miR-197	Wnt/β-catenin, Axin-2, NKD1 and DKK2	Metastasis	[86]
	miR-219-5p	cadherin 1	Tumor growth/Metatasis	[87]
	miR-221 and miR-222	PHF2, AKT pathway, PTEN, CDK inhibitor p27 and DDIT4	Occurrence/Development/ metastasis	[88] [89]
	miR-224	CDC42, CDH1, PAK2, BCL-2, MAPK1, SMAD4 and HOXD10, PPP2R1B, AKT-signaling	Proliferation/Invasion/ Migration/HCV-induced HCC	[63] [64]
	miR-452	Sox7, Wnt/β-catenin signaling	CSC in HCC	[59]
	miR-487a	PIK3R1-AKTsignaling SPRED2-protein kinase signaling	Proliferation/Metastasis	[90]
	miR-494-3p	Focal adhesion and MAPK signaling	Invasion/Metastasis	[71]
	miR-500a	BID, SFRP2 and GSK-3β, and Wnt/β-catenin signaling	Progression/Migration/ Invasion	[91]
	miR10b, miR147, miR338, miR-597	HOXD10, TPM1, TRAIL3, NOL3, ZAP-70, SMAD2, LODH1, IRF2 and KLF3	Recurrence after liver transplant	[67]
	miR-1269	FOXO1, p21, cyclinD1	Proliferation	[58]

**Table 2 biomolecules-12-00645-t002:** Diagnostic and prognostic miRs in HCC.

miRNA	Expression	Observed Location	Type of HCC	References
miR-718	Upregulated	Serum Exosome	Biomarker of HCC	[134]
exo-miR-18a, exo-miR-221, exo-miR-222 and exo-miR-224	Upregulated	Serum Exosome	Biomarker of HCC	[135]
exo-miR-215-5p	Upregulated	Serum Exosome	Predictive biomarker of HCC	[136]
exo-miR-122a, exo-miR-21 and exo-miR-96	Upregulated	Serum Exosome and plasma	Biomarker for the early identification of HCC	[137]
exo-miR-92b	Upregulated	Serum Exosome	Predicting the probability of HCC recurrance	[139]
(a) miR-221, miR-222, miR-93, miR-21 and miR-224 (b) miR-130a, miR-195, miR-199a and miR-375	Upregulated Downregulated	------	Diagnosis and therapeutics targest to malignancy	[142]
(a) miR-1269, miR-224, miR-224-3p and miR-452 (b) miR-199a-5p, miR-199a-3p and miR-199b	Upregulated Downregulated	------	HCV-induced HCC	[143]
miR-122, miR-199a and miR-16	Upregulated	Serum	Biomarkers of HCV-induced HCC	[144,145]
miR-375, miR-92a, miR-10a, miR-223, miR-423, miR-23b, miR-23a, miR-342-3p, miR-99a, miR-122a, miR-125b and miR-150	Upregulated	Serum	noninvasive biomarker in HBV-mediated HCC	[146]
miR-92-3p, miR-107 and miR-3126-5p	Upregulated Downregulated	Serum	Early stage and low-level AFP-HCC	[147]
miR-20a-5p, miR-25-3p, miR-30a-5p, miR-92a-3p, miR-132-3p, miR-185-5p, miR-320a and miR-324-3p	Upregulated	Serum and tissue	Diagnosis of HBV-positive HCC	[148]
miR-192-5p, miR-21-5p and miR-375	Upregulated	Serum and tissue	Biomarker for HCC screening	[149]
miR-192	Upregulated	Serum Exosome	Biomarker of HCC	[150]
miR-665	Upregulated	Serum	Diagnosis of HCC	[151]
miR-148a	Downregulated	Serum	Prognosis of HCC	[152]
miR-126	Upregulated	Plasma	HBV HCC	[153]
miR-424	Downregulated	Serum	Prognosis of HCC	[154]

**Table 3 biomolecules-12-00645-t003:** miRs in HCC therapeutics.

Therapeutic Strategy	miRNA	Delivery System	Mouse Model	References
miRNA	miR-221	Anti-miR-221-AMO	Orthotopic model	[181]
	miR-487a	Morpholino-anti-miRNA miR-487a	Orthotopic model	[90]
	miR-500a-3p	antagomiR-500a-3p	Xenograft	[91]
	miR-122	Miravirsen, anti-mir-122	Phase III Clinical	[180]
	miR-21	anti-miR-21	Xenograft	[182]
	miR-494	miR-494-anti-miR	tet-o-MYC; LAP-tTA	[176]
miRNA replacement therapy	miR-26a	AAV vector	tet-o-MYC; LAP-tTA	[41]
	miR-29b	2’O-methyl modified miRNA	Xenograft	[183]
	miR-206	Plenti_CMV-puromycin- miR206	Xenograft	[76]
	miR-302b	Lipofectamine RNAi-MAX	Xenograft	[42]
	miR-361-5p	oligonucleoides miR-361-5p mimics in Lentivirus	Xenograft	[78]
	miR-101	Lipofectamine RNAi-MAX	Xenograft	[68]
	miR-199a	oligonucleoides Lenti-miR-199a	Xenograft	[184]
	miR-375	Cholesterol-conjugated 2’O-methyl modified Oligo mimics	Xenograft	[47]
	miR-122	AAV	tet-o-MYC; LAP-tTA	[185]
	miR-422a	Oligo mimics; Lentivirus	Xenograft; DEN-induced mouse model	[186]
	miR-21	Antisense MiR-21 co- encapsulated with gemci- tabine in PEGylated-PL GA nanoparticles	in vitro only	[187]
	miR-375	AuNP-miR-375	Xenograft	[188]
	miR-122	Cationic lipid nanoparticles	Xenograft	[189]
	miR-34	Liposome-based miR-34 mimic (MRX34)	Phase I clinical trial	[190]
	miR-124	Liposomes	DEN-induced HCC mouse model	[191]
Oncolytic Virus	let-7	Adenovirus	Xenograft	[192]
	miR-122	Adenovirus	Xenograft	[193]
	miR-122; let-7 miR-124	Herpes Simplex virus	Xenograft	[194]
	miR-199	Adenovirus	Xenograft; TG-221	[195]

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
