# Peer review of "Hepatocellular Carcinoma: The Role of MicroRNAs"

_biomolecules, 2022, doi:10.3390/biom12050645_

Round 1

Reviewer 1 Report

The manuscript entitled “Hepatocellular Carcinoma: Role of MicroRNAs” is a review paper focused on the roles of miRNAs in hepatocellular carcinoma progression, metastasis, and treatment. The manuscript can be of interest to the journal audience. However, there are some concerns and recommendations.

  1. The manuscript is not well-organized. There are many repeats and inconsistencies.
  2. The roles of miRNAs in HCC heterogeneity are not clearly described. Moreover, the phenomenon of tumor heterogeneity is not properly discussed. What was a reason to include section 2.2 “Genetic heterogeneity in HCC” if differentially expressed genes, driver mutations, and gene amplification were not discussed?
  3. The term “epigenetic” implies, first of all, at least among others, DNA methylation and histone modification. Therefore, the title of section 2.3 should be re-considered.
  4. Table 1 should be reconsidered since each miR can regulate a definite signaling pathway. Additionally, Refs should be checked because some papers are incorrectly cited.
  5. In section 4, the roles of miRNAs for differential diagnostics of HCC are discussed. Prognostic values of miRNAs are not properly discussed, instead, some discussion of predictive value in HCC treatment is given.
  6. In section 5, strategies to block or to replace miRNAs, but not to use them for HCC therapy are discussed. Therefore, the title of this section should be changed.
  7. More Figures are recommended. Also, there are no Refs. For the last two years discussed.

Reviewer 2 Report

This manuscript (biomolecules-1606745) revised extensively the role and potential of microRNAs in pathogenesis and to be diagnostic and prognostic biomarkers or therapeutical targets of hepatocellular carcinoma. The theme is interesting and very current, due to its clinical potential in a disease with a high death rate and few successful therapeutic strategies. The review was written clearly and objectively. I would like to mention just a few considerations:

  1. It would be interesting that the authors do a table with the information mentioned in section 4 (Diagnostic and Prognostic microRNAs in HCC), similar to table 1 or 2, showing the microRNA, the expression change, the type of fluid (serum, plasma, or other), the serum circulation way (free, inside extracellular vesicles, coupled to proteins, etc) and the type of HCC (HCV-induced HCC, NAFLD-induced HCC, unknown, etc).

  1. The authors only mentioned one turn the miRNAs-containing exosomes as promising biomarkers for HCC recurrence after liver transplantation (line 241). However, serum miRNAs-containing extracellular vesicles have been emerged as promising biomarkers, but mainly as important cell-to-cell messengers standing out their potential role in cancer metastasis.  The authors should enumerate the current scientific knowledge about this and discuss their potential in HCC.

  1. The authors mentioned that a mimic of miR34 (MRX34) has been used in a phase 1 clinical trial (line 329), but do not refer that the company that was to develop it, halts the study due to multiple immune-related severe adverse events over the course of the trial (https://www.businesswire.com/news/home/20160920006814/en/Mirna-Therapeutics-Halts-Phase-1-Clinical-Study-of-MRX34). The authors should refer and discuss the implications of this event to the use of miRNAs as therapeutic strategy to treat HCC.

Round 2

Reviewer 1 Report

The manuscript entitled “Hepatocellular Carcinoma: Role of MicroRNAs” is a review paper focused on the roles of miRNAs in hepatocellular carcinoma progression, metastasis, and treatment. The manuscript has been substantially revised. Most my recommendations and and concerns have been addressed.